# Understanding molecular mechanisms and predicting phenotypic effects of pathogenic tubulin mutations

**Thomas J. Attard[1], Julie P. I. Welburn[1]\*, Joseph A. Marsh[2]\***

**1** Wellcome Trust Centre for Cell Biology, School of Biological Sciences, University of Edinburgh, Edinburgh, Scotland, United Kingdom, **2** MRC Human Genetics Unit, Institute of Genetics & Cancer, University of Edinburgh, Edinburgh, United Kingdom

\* julie.welburn@ed.ac.uk (JPIW); joseph.marsh@ed.ac.uk (JAM)

## Abstract

Cells rely heavily on microtubules for several processes, including cell division and molecular trafficking. Mutations in the different tubulin-α and -β proteins that comprise microtubules have been associated with various diseases and are often dominant, sporadic and congenital. While the earliest reported tubulin mutations affect neurodevelopment, mutations are also associated with other disorders such as bleeding disorders and infertility. We performed a systematic survey of tubulin mutations across all isotypes in order to improve our understanding of how they cause disease, and increase our ability to predict their phenotypic effects. Both protein structural analyses and computational variant effect predictors were very limited in their utility for differentiating between pathogenic and benign mutations. This was even worse for those genes associated with non-neurodevelopmental disorders. We selected tubulin-α and -β disease mutations that were most poorly predicted for experimental characterisation. These mutants co-localise to the mitotic spindle in HeLa cells, suggesting they may exert dominant-negative effects by altering microtubule properties. Our results show that tubulin mutations represent a blind spot for current computational approaches, being much more poorly predicted than mutations in most human disease genes. We suggest that this is likely due to their strong association with dominant-negative and gain-of-function mechanisms.

## Author summary

Filament-like structures, called microtubules, are essential for cells to function, distribute material around the cell and organisms, and help cells grow. The building blocks of microtubules are proteins called tubulins, which can rapidly polymerise and depolymerise. Mutations in tubulin genes can have catastrophic consequences on many different types of cells, leading to diseases such as bleeding defects, female infertility, and disorders impairing brain development. However, how these mutations cause disease and whether they can be predicted is still unknown. We used computational and experimental techniques to address these issues. First, we compared how disease-causing tubulin mutations

**Data Availability Statement:** All relevant data are provided within the manuscript and its Supporting Information files.

**Funding:** This work was supported by the Medical Research Council via a Precision Medicine Doctoral

Training Programme studentship to TJA and a Career Development Award (MR/M02122X/1) to JAM. JAM is a Lister Institute Research Fellow. JPIW is supported by a Wellcome Trust Senior Research Fellowship (207430). The funders had no role in study design, data collection and analysis, decision to publish, or preparation of the manuscript.

**Competing interests:** The authors have declared that no competing interests exist.

and ones found in healthy people impact the structure of tubulin. Then, we tested the ability of available computational predictors to distinguish between these two types of tubulin mutations. We found these programs poorly predict tubulin mutations that cause diseases, limiting their usefulness. Next, we studied disease-causing mutations that were not predicted by computational methods. We found that these did not prevent tubulin from forming microtubules, indicating these mutations change the function of tubulin without inactivating them. Our work presents tubulins as a weakness of current computational predictors, potentially because they fail to consider different ways in which mutations cause disease.

## Introduction

Microtubules are polarised cytoskeletal filaments essential in several cellular processes, ranging from cell division to signalling and transport. They assemble into axons, cilia, and the mitotic spindle, while also providing tracks for microtubule-associated proteins (MAPs) and motors for molecular trafficking [1,2].

Microtubules self-assemble from tubulin-α and -β heterodimers, with the dynamics of their assembly and disassembly being integral to their function [3]. Tubulin-α and -β are ubiquitous in eukaryotes, while related proteins in the FtsZ family show similarity in sequence, structure and function in archaea and bacteria [4,5]. Nine tubulin-α and ten tubulin-β genes have been identified in humans, originating from evolutionary gene duplication events [6]. In the tubulin field, these tubulin paralogues are referred to as isotypes. Between tubulin-α and -β, conservation in sequence and structure is high, especially at interfaces stabilising the heterodimer, and at contacts between tubulin heterodimers across (lateral) and along (longitudinal) protofilaments [7]. While tubulin-α and -β both bind to GTP, only tubulin-β can hydrolyse it to GDP, with residues on these binding sites amongst the most conserved [8]. GTP hydrolysis enables distinct conformations that mediate the dimer's ability to be incorporated into microtubules [9]. The C-terminal region makes up the outer surface of the microtubule and so contributes to most of the interactions with MAPs [10]. Furthermore, many differences in amino acid sequences between isotypes occur in this region [11] and in the unstructured, highly negative tail [12]. Tubulin-γ, δ and ε are more divergent in sequence than tubulin-α and -β and are involved in the basal bodies of centrioles, rather than being self-assembled into dynamic polymers [13,14].

A wide range of genetic disorders–called 'tubulinopathies'–have now been attributed to tubulin mutations. Over 225 pathogenic mutations in human tubulin isotypes have been reported (see Table 1). These findings highlight the importance of understanding tubulin function in different cell types. The first reported tubulin mutations associated with pathogenic phenotypes were found in TUBα1A, TUBβ2B, TUBβ3, and TUBβ4A and resulted in neurodevelopmental defects [15]. Mutations in the γ-tubulin isotype TUBγ1 –which is necessary for microtubule nucleation–also cause a similar neurodevelopmental disorder [16,17]. While mutations in TUBα4A and TUBβ4A have been linked to neurodegenerative disease [18,19], phenotypes outside the nervous system are now emerging. These include TUBβ1 mutations associated with bleeding disorders [20], a link between tubulin-α acetylation and reduced sperm motility [21], and TUBβ8 mutations connected with female infertility due to incorrect meiotic spindle assembly [22].

Despite the large number of identified pathogenic tubulin mutations, our understanding of the molecular mechanisms by which these mutations cause disease remains limited. The large

**Table 1. Summary of all tubulin isotypes considered for this study.** The number of dominant pathogenic mutations identified for each isotype is denoted, as well as their pathogenicity classification. Where necessary, additional comments about the pathogenicity type and references for all mutations are also included. MCM = missense constraint metric (Z-Score obtained from gnomAD); ALS = amyotrophic lateral sclerosis; H-ABC = hypomyelination with atrophy of basal ganglia and cerebellum.

| Isotype | # pathogenic | # gnomAD | MCM | Phenotype classification | References and comments |
|---|---|---|---|---|---|
| TUBα1A | 67 | 680 | 5.584 | neurodevelopmental | Various neuropathies [35,79–83] |
| TUBα1B | 0 | 696 | 5.412 | cancer-only | Roles in resistance to chemotherapy [84–86] |
| TUBα1C | 0 | 153 | 2.171 | cancer-only | Identified as an oncogene [87] |
| TUBα3D | 0 | 343 | 1.533 | non-disease | |
| TUBα3E | 0 | 531 | -0.296 | non-disease | Only 1 homozygous mutation known, which was excluded [88] |
| TUBα4A | 7 | 255 | 3.303 | other (neurodegenerative) | All linked with ALS [19]. One deletion mutation also identified (excluded) |
| TUBα8 | 0 | 201 | 0.631 | non-disease | Homozygous 14bp deletion linked with polymicroglia [24] (excluded) |
| TUBβ1 | 6 | 268 | 0.003 | other (bleeding disorders) | Linked to platelet defects [45–47] |
| TUBβ2A | 3 | 192 | 5.263 | neurodevelopmental | Simplified gyral patterning, infantile-onset epilepsy and progressive spastic ataxia [30, 89] |
| TUBβ2B | 29 | 190 | 5.120 | neurodevelopmental | Various neuropathies [15,29,55,57,64,90–92] |
| TUBβ3 | 24 | 76 | 4.579 | neurodevelopmental | Various neuropathies, a few mutations associated with milder phenotypes [15,32,55,57,64,66,93,94] |
| TUBβ4A | 38 | 198 | 4.262 | neurodevelopmental | Mostly H-ABC and Hypomyelination, some less severe and a few linked with Dystonia [56,95–112] |
| TUBβ4B | 2 | 201 | 4.498 | other (sensory) | Linked heavily with cancer [113]. Two mutations have been linked to a sensorineural neurodegenerative disease impairing vision [18] |
| TUBβ5 | 7 | 198 | 5.625 | neurodevelopmental | Various neuropathies, mostly severe [28,114–119] |
| TUBβ6 | 1 | 74 | 2.637 | neurodevelopmental | Linked with non-progressive congenital facial palsy [120] |
| TUBβ8 | 35 | 464 | 1.860 | other (female infertility) | Linked to oocyte maturation defects [22,23,50–53] |
| TUBβ8B | 0 | 228 | -2.372 | non-disease | |
| TUBγ1 | 8 | 57 | 4.155 | neurodevelopmental | All linked to cortical development malformations, one of unknown inheritance [16,17] (excluded). Two deletions also cause a lack of centrosome localisation [121] |
| TUBγ2 | 0 | 113 | 2.443 | non-disease | |

majority of pathogenic tubulin mutations involve missense changes (*i.e.* single amino acid residue substitutions) and have autosomal dominant inheritance. There are only a few known exceptions, including homozygous null and in-frame deletion mutations in TUBβ8 that can cause an oocyte maturation defect [23], a dominant nonsense mutation that results in a slightly truncated TUBα4A linked to the neurodegenerative disease amyotrophic lateral sclerosis [19], and a recessive intron deletion in TUBα8 in polymicrogyria patients that interferes with splicing, producing shorter mRNAs that do not contain exon 2 [24].

The absence of any known protein null mutations in tubulins causing dominant disease is striking. This tells us that the molecular mechanism underlying dominant mutations cannot simply be haploinsufficiency, whereby disease is caused by a complete lack of functional protein produced from the mutant allele. One possibility is that disease is caused by a milder loss of function (*i.e.* hypomorphic mutations). For example, pathogenic TUBα1A mutations have been reported to disrupt interactions of the nascent protein with tubulin chaperones, which impairs heterodimer formation, suggesting that disease can be caused by a partial loss-of-function [25]. Alternatively, a pathogenic mutation could act via a non-loss-of-function mechanism, having a dominant-negative effect or causing a gain of function [26]. This would typically be associated with the mutant protein retaining the ability to incorporate into microtubules, which has been observed for many pathogenic tubulin mutations [27–32]. For dominant-negative mutations, the incorporation of mutant protein directly or indirectly disrupts

the activity of the wild-type protein [26]. In these cases, the mutant tubulin retains its ability to form a heterodimer and assemble into microtubules before consequently impacting function in some other way, *e.g.* by perturbing microtubule properties or disrupting interactions with MAPs. For instance, TUBβ3 mutations that alter the charged surface of the microtubule prevent molecular motors from binding and thus have profound impacts on cellular transport [33]. Changes in microtubule properties could also induce a gain of function, as has been proposed for the pathogenic T178M variant in TUBβ2A and TUBβ3, which has been reported to make microtubules more stable and cause altered microtubule growth dynamics [34].

With the increasing accessibility of sequencing data, novel tubulin variants are being continually discovered [35]. Since it is impractical to test them all experimentally, there is a strong need for computational approaches to identify tubulin mutations most likely to be pathogenic. Many variant effect predictors (VEPs) have been developed in recent years, and some of these are now in widespread use to help identify mutations that potentially have clinical significance [36]. However, the performance of these predictors can vary quite dramatically across different proteins and, to our knowledge, there has been no systematic assessment of their performance on tubulin mutations specifically. Missense mutations that cause pathogenicity by dominant-negative or gain-of-function mechanisms tend to be poorly predicted by most currently available VEPs [37], which could potentially limit their applicability to tubulins. It is therefore important for us to understand to what extent we can rely on computational predictors when assessing tubulin mutations.

In this study, we have first performed a systematic survey of known pathogenic missense mutations across all human tubulins and analysed their positions within the three-dimensional structures of tubulin heterodimers. This approach has allowed us to look for patterns in mutations across isotypes, structural locations, and phenotypes, in an attempt to obtain insight into the likely molecular disease mechanisms. Next, we assessed the performance of several different VEPs in distinguishing between pathogenic and putatively benign missense variants, observing that the predictive performance of all tested methods is poor compared to most other proteins. Finally, we have selected pathogenic tubulin mutations that were poorly predicted by computational approaches for experimental characterisation and show that the mutant proteins are able to be incorporated into microtubules, consistent with a likely dominant-negative mechanism. Our work suggests that many tubulin pathogenic mutations that act via non-loss-of-function mechanisms cause pathogenic phenotypes that cannot be explained computationally using current methods that rely on sequence conservation or protein structure. Overall, this study highlights the need for a greater understanding of microtubule-protein interactions to understand the molecular mechanisms underlying tubulinopathies.

## Results

### Survey of tubulin missense mutations

First, we compiled as many previously identified pathogenic or likely pathogenic dominant tubulin missense mutations as possible, using online databases [38,39] and extensive literature searching (S1 Table). In addition, we also identified missense variants in tubulin genes observed across >140,000 people from the gnomAD v2.1 database [40]. Given that the gnomAD dataset comprises mostly healthy individuals without severe genetic disorders, these variants are unlikely to cause dominant disease, and we therefore refer to them as "putatively benign". However, we acknowledge that some of these variants could have milder effects, variable penetrance, or be associated with late-onset disease. Table 1 shows the numbers of pathogenic and gnomAD missense variants for each tubulin isotype and the associated type of

genetic disease. Somatic mutations in tubulins are also implicated in cancer development; however, these mutations are likely to provide a selective advantage to cancer cells by providing resistance to chemotherapeutic drugs [41–43]. Hence these mutations might obscure our results and have not been included in our study, although we have noted two isotypes with links to cancer in which no other disease-related mutations have been identified yet (Table 1). While pathogenic mutations occur throughout the tubulins, mutations in both tubulin-α and -β show clustering towards the intermediate and C-terminal domains when shown in the context of the linear amino acid sequence (Fig A in S1 Text).

We also considered the gene-level missense constraint metric (MCM) scores provided by the gnomAD database (Table 1). These are derived from a model based on sequence context to predict the number of expected variants present in a healthy population relative to the number of actual variants observed [44]. They provide a metric for the tolerance of each isotype to missense variation, with higher values representing genes that are more intolerant to amino acid sequence changes. Interestingly, we observed high MCM scores for isotypes linked with neurodevelopmental disorders (TUBα1A, TUBβ2A, TUBβ2B, TUBβ3, TUBβ4A and TUBβ5). TUBβ6 is the only exception, and has only one pathogenic mutation reported so far, causing congenital facial palsy. These scores contrast with the much lower scores observed for TUBβ1 and TUBβ8, associated with platelet defects [45–49] and female infertility [22,23,50–53], respectively, which suggest that they are much more tolerant to sequence variation. Overall, our analysis indicates there is a stronger sequence constraint in the human population for tubulin isotypes that contribute to neurodevelopment. This may be due to the selective pressure of the process, compared to tubulin isotypes expressed in cells that affect organism fitness to a lesser extent.

## Protein structural context is of limited utility for explaining tubulin mutation pathogenicity

Next, we mapped all variants to three-dimensional structures of tubulins in the Protein Data Bank (PDB). We classified each mutation-carrying residue from every tubulin isotype based upon its structural location: whether it occurs at an intermolecular interface, on the protein surface, or buried in the protein interior. We used a hierarchy to classify structural locations (Fig B in S1 Text). First, given that GTP binding (and hydrolysis for tubulin-β) is essential, we classified residues on the GTP nucleotide binding site as 'GTP binding'. Other tubulin mutations occurring at intermolecular interfaces were classified into three categories depending on where they occur in the model structures described in S2 Table: at the intradimer interface, at an interdimeric interface (which we refer to as microtubule contacts, as they support lateral and longitudinal interactions with microtubules), or at interfaces with MAPs. Remaining mutations not at interface positions were then classified depending on whether they were on the protein interior or surface according to relative solvent accessible surface area [54]. Finally, we labelled any residues that were not present in any ordered part of any experimentally derived tubulin structures as 'outside of structure'. These typically occurred in the disordered C-terminal region, or at positions of divergence between the sequences of mapped structures and the tubulin isotype in question. Notably, while several gnomAD variants occurred at 'outside of structure' positions, this was true for only a single pathogenic mutation: R307H in TUBβ1. Interestingly, all other tubulin-β isotypes have a histidine at this position, explaining why the residue could not be mapped within our pipeline, as the wild-type amino acid residue needs to be present in the structure.

First, we used this classification system to assess whether tubulin mutations are enriched at particular locations. Despite the large dataset size, no significant differences in the locations of

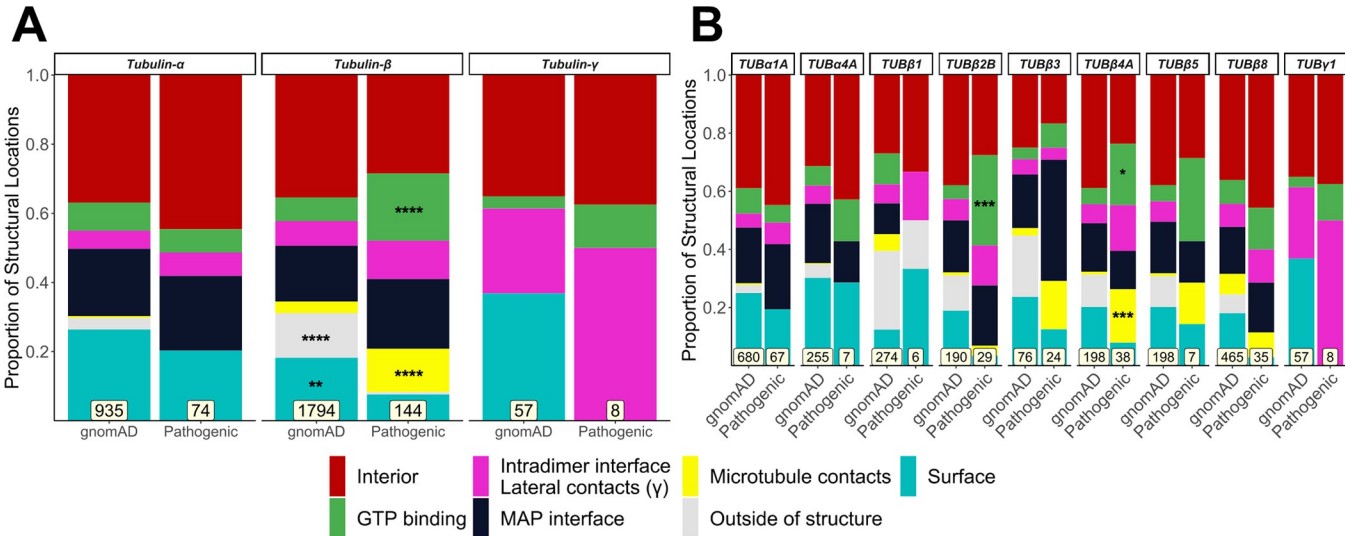

**Fig 1. Comparison of the distribution of structural locations between pathogenic and putatively benign tubulin variants.** The proportion of pathogenic and putatively benign gnomAD mutations in each location type for tubulin-α, β and γ globally (**A**) and for individual isotypes (**B**). Mutation totals for each group are shown at the bottom, and only isotypes with at least five pathogenic mutations were included. Fisher's exact tests were used to compare frequencies between gnomAD and pathogenic mutations considering each family and isotype separately. Asterisks indicate a location class with a significantly higher proportion of mutations compared to its corresponding group, where * $p < 0.05$, ** $p < 0.01$, *** $p < 0.001$, **** $p < 0.0001$.

pathogenic or putatively benign gnomAD mutations were found across tubulin-α isotypes collectively (Fig 1A) or in any tubulin-α isotype individually (Fig 1B). This result was surprising given that the link between TUBα1A and disease remains the most well-established of all tubulin isotypes, with 67 pathogenic variants linked with various neurodevelopmental defects [35]. Observations were similar for TUBα4A, the only other tubulin-α isotype with dominant pathogenic mutations (Fig 1B) [19]. Furthermore, pathogenic mutations in both isotypes showed no obvious patterns or spatial clustering when visualised within the protein structure (Figs 2 and C and D in S1 Text).

In contrast to tubulin-α, we do observe a significant enrichment of pathogenic tubulin-β mutations relative to gnomAD variants at GTP binding residues and at lateral or longitudinal (*i.e.* interdimeric) microtubule contacts (Fig 1A), especially in TUBβ2B and TUBβ4A (Fig 1B). Mutations in these isotypes have been extensively linked to neuronal defects, and can be observed to form spatial clusters on tubulin structures (Fig 2). In contrast, putatively benign gnomAD variants are significantly more likely to occur on the protein surface or at 'outside of structure' positions (Fig 1B). TUBβ3 and TUBβ5 are also linked to similar disease phenotypes, but do not show any notable differences in structural location [15,28,55–57].

Interestingly, we found no significant differences in structural location in tubulin-β isotypes associated with diseases outside the neuronal system (Fig 1B), nor did pathogenic mutations show any clear patterns or clustering at the three-dimensional structure level (Fig 2). We did, however, note that many TUBβ1 variants occurred at 'outside of structure' residues (Fig 1B). This is due primarily to the greater sequence divergence between TUBβ1 and experimentally determined tubulin structures, as any positions where the TUBβ1 sequence is different than the aligned positions within the available structures will remain unmapped using our approach.

The centrosome-localised TUBγ1 has also been associated with neurodevelopmental disease [16,17]. Using the crystal structure of TUBγ1 containing two molecules per asymmetric unit (PDB ID: 3CB2) [58], we observed that 4 out of 8 pathogenic mutations identified so far occur at interfaces responsible for lateral interactions between TUBγ1 monomers (Figs 1 and

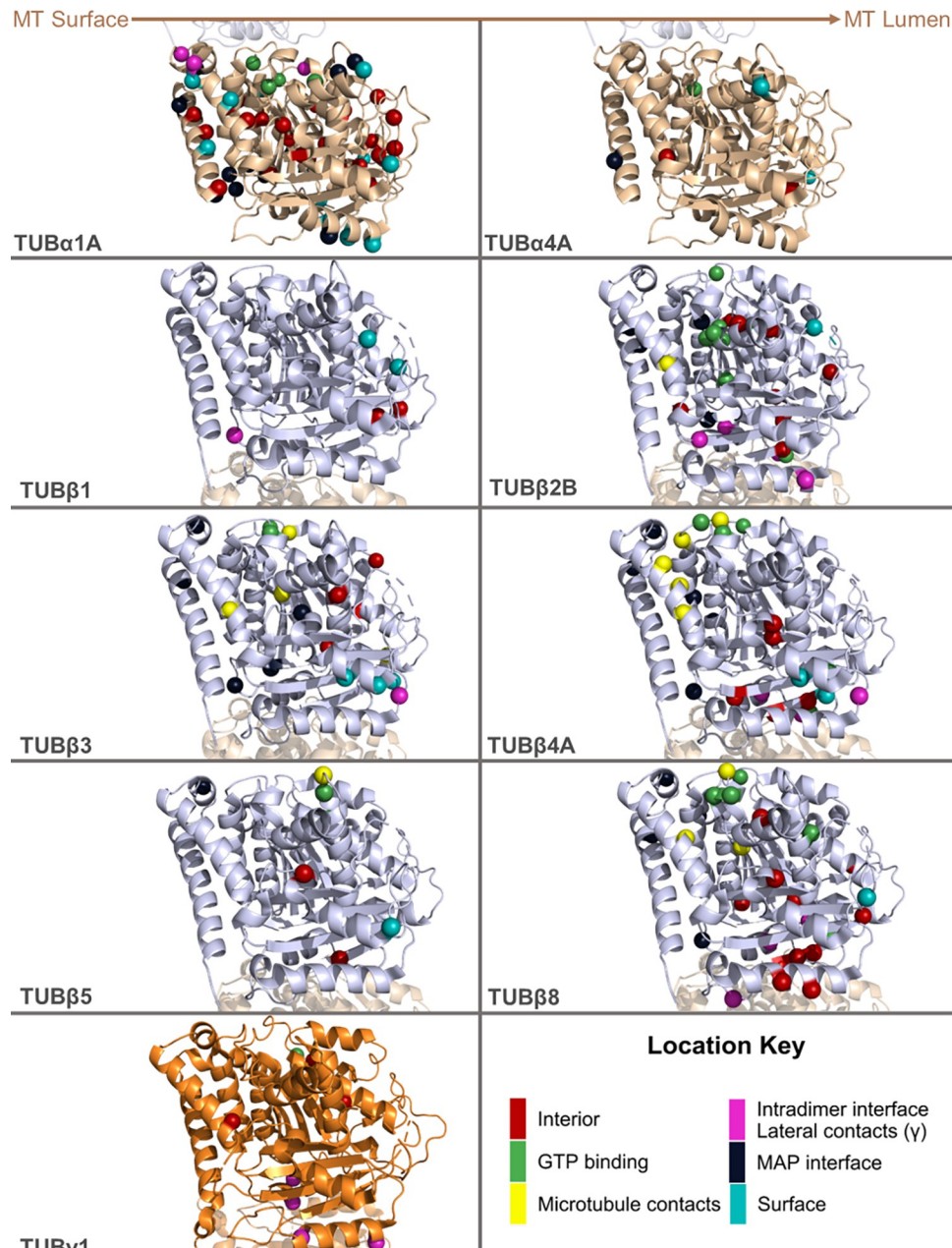

**Fig 2. Visualisation of pathogenic tubulin mutations on three-dimensional structures of tubulin heterodimers.**
Structures of tubulin isotypes with at least five identified pathogenic mutations. Coloured residues indicate pathogenic mutations according to their location. The position of the tubulin subunit in relation to the microtubule (MT) is noted at the top. Residues in red denote mutations buried in the protein interior, while light cyan residues are on the surface. Green residues indicate mutations occurring at GTP binding interfaces, while ones in magenta highlight residues on intradimer interfaces. Navy blue and yellow residues mark mutations on residues interacting with microtubule associating proteins (MAPs) and other tubulin dimers, respectively. PDB IDs: 6s8k (for all tubulin-α and β isotypes) and 3cb2 (for TUBγ1). Alternate versions of this figure with the structures rotated 90˚ and 180˚ around the y-axis are provided in Figs C and D in S1 Text.

2, labelled in magenta). We suspect these interactions to occur between adjacent TUBγ1 chains within γ-TURC [59]. Their distribution was not significantly different to gnomAD variants, although we note the small sample size.

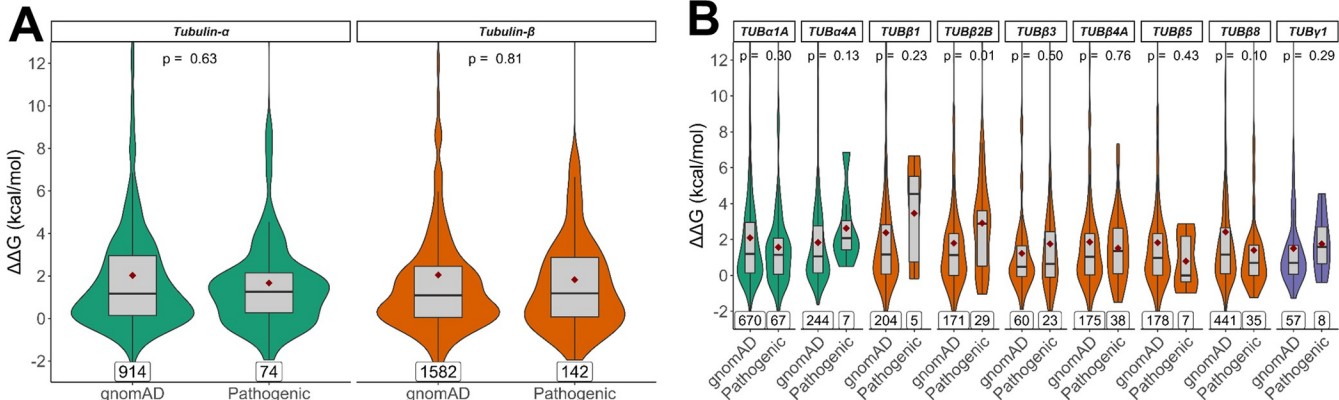

**Fig 3. Comparison of predicted changes in protein stability between pathogenic and putatively benign tubulin mutaitons.** ΔΔG values representing the predicted change in free energy of folding were calculated with FoldX considering the structure of the monomeric subunit only. Scores are shown for tubulin-α and -β families globally (**A**) and in isotypes with at least five identified pathogenic mutations (**B**). Maroon diamonds indicate the mean ΔΔG values, and mutation totals for each group are also shown at the bottom. The p-values displayed were obtained via unpaired Wilcoxon tests.

## Most pathogenic tubulin mutations are not highly disruptive to protein structure

Next, we considered the predicted structural perturbations of pathogenic and putatively benign gnomAD missense variants using FoldX [60]. This outputs a ΔΔG value, in units of kcal/mol, with positive values indicating that a mutation is likely to destabilise protein structure and negative values indicating predicted stabilisation. Previous work has shown that computationally predicted ΔΔG values can sometimes show considerable utility for the identification of pathogenic missense mutations, and for understanding likely molecular disease mechanisms [61].

Interestingly, we observe no significant differences between the ΔΔG values of pathogenic and gnomAD missense variants for tubulin-α, -β or -γ (Fig 3A). Of the individual isotypes, only TUBβ2B shows significantly higher ΔΔG values for pathogenic mutations (p = 0.01), although this would not remain significant when accounting for multiple testing (Fig 3B). We initially used ΔΔG values that only consider the structural impact of variants on the monomer alone, as they are more consistent between structures. However, we also observed very similar results using the full ΔΔG values calculated using the entire complex, including intermolecular interactions, as well as when using absolute ΔΔG values (Fig E in S1 Text).

These results suggest that the structural destabilisation is not a primary molecular disease mechanism underlying pathogenic tubulin mutations, and that considering structural impact is not particularly useful for differentiating between pathogenic and benign tubulin variants. Notably, this aligns with recent work showing that the predicted effects on protein stability tend to be much milder in gain-of-function and dominant-negative mutations than for pathogenic mutations associated with a loss of function [37], supporting the idea that most pathogenic tubulin mutations are due to non-loss-of-function effects.

## Variant effect predictors show poor performance in discrimination between pathogenic and putatively benign tubulin mutations

Next, we assessed the abilities of 25 different VEPs to distinguish between pathogenic and putatively benign tubulin missense mutations. A complete set of predictions for every tubulin mutation from all VEPs is provided in S3 Table. To compare the performance of different

VEPs, we first used a metric of predictor performance, known as the receiver operating characteristic (ROC) area under the curve (AUC) generated from each VEP over the entire dataset of tubulin missense variants (Fig 4A). Overall, the VEPs performed very poorly. Most had overall AUCs below 0.6, with the top-performing predictor, REVEL, having an AUC of only 0.68. In contrast, a recent study using a very similar methodology found that many VEPs had overall AUCs above 0.8, e.g. REVEL had an AUC of 0.9 for haploinsufficient disease genes, 0.85 for genes associated with a gain-of-function, and 0.83 for genes associated with dominant-negative effects [37]. Thus, even considering that VEPs tend to do worse for non-loss-of-function mutations, the performance we observe here for tubulin mutations is strikingly poor.

It is important to note that our analysis is likely to overstate the predictive power of some VEPs. Supervised machine learning approaches underpin most VEPs, and typically use datasets of known pathogenic and benign variants for training. Since some VEPs are likely to have been trained using some of the tubulin mutations in our evaluation, their performance has a strong possibility of being overstated. This problem is particularly acute for metapredictors, including the top-performing methods in our analysis, REVEL and M-CAP, which combine supervised learning with multiple other predictors as inputs. In contrast, predictors based upon unsupervised machine approaches and those utilising empirical calculations should be free from this bias. Therefore, given the performance of the unsupervised predictor DeepSequence, ranking third overall, we likely consider it to be the most reliable predictor of tubulin mutation pathogenicity, consistent with its top-ranking performance in a recent study [62]. However, even DeepSequence only achieves an AUC of 0.63 for tubulin mutations here, compared to well over 0.8 for all disease-associated proteins tested in that study.

Next, we compared the AUCs calculated for individual isotypes across all VEPs, considering isotypes with at least 10 pathogenic mutations. We found that pathogenic mutations in the tubulin-β isotypes TUBβ2B, TUBβ3 and TUBβ4A, which are all associated with neurodevelopmental diseases, were predicted better compared to TUBβ8, associated with oocyte maturation defects (Fig 4B). Therefore, we classified tubulins into two groups based upon observed disease phenotypes: neurodevelopmental and other (as classified in Table 1). We observed a significantly better performance on isotypes linked to neurodevelopmental diseases (Fig 5A).

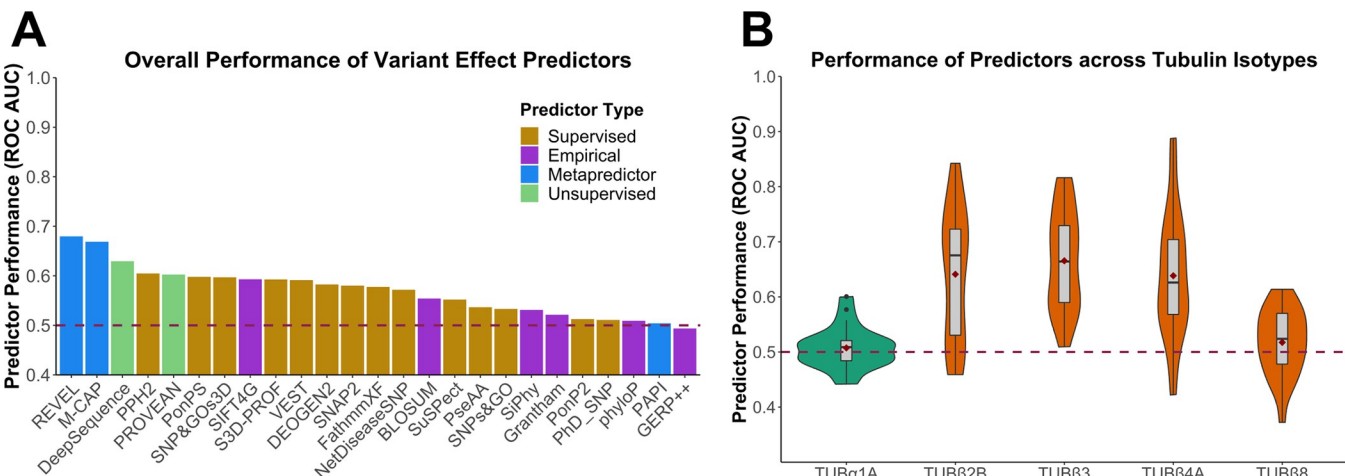

**Fig 4. Assessment of VEP performance for identification of pathogenic tubulin mutations. (A)** ROC AUC values for each VEP across all tubulin-α and -β isotypes with at least one identified pathogenic mutation, colour coded according to predictor category. **(B)** Distribution of ROC AUC values across all VEPs for isotypes with at least 10 identified pathogenic missense mutations. Dashed line indicates the performance of a random predictor. Maroon diamonds indicate the mean area.

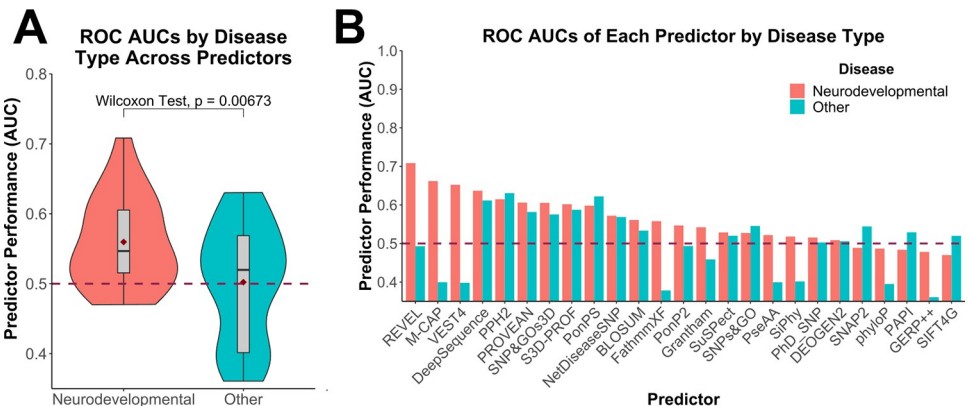

**Fig 5. Comparison of VEP performance on pathogenic mutations in tubulin genes associated with neurodevelopmental vs other disease phenotypes. (A)** Distribution of ROC AUC values across all VEPs for isotypes with mutations linked with neurodevelopmental or other disease phenotypes. The p-value stated was obtained via a paired Wilcoxon test. Maroon diamonds indicate the mean area. **(B)** ROC AUC values for each VEP in isotypes with mutations associated with neurodevelopmental or other disease phenotypes. Dashed line indicates the performance of a random predictor.

Interestingly, when we compare the performance on neurodevelopmental *vs* other disorders across the individual VEPs (Fig 5B), we observe that metapredictors and supervised VEPs, like REVEL, M-CAP and VEST4, outperformed all other VEPs on isotypes linked to neurodevelopmental disease but showed a drastic decrease in performance on other phenotypes. In contrast, unsupervised DeepSequence shows very similar performance between the two groups. This strongly suggests that that certain VEPs have likely been overfitted in their training against the neurodevelopmental mutations.

Given its overall performance in our analyses and unsupervised nature, we currently recommend DeepSequence for predicting the effects of tubulin mutations, although we emphasise that its predictive utility is still relatively limited. Therefore, we have produced DeepSequence predictions for every possible amino acid substitution across most tubulin isotypes and have provided them as a resource (S4 Table). We have also calculated optimal thresholds for DeepSequence using the closest point to the top left corner of our ROC curves. Based upon this, we suggest that DeepSequence scores lower than -5.89 are likely to be pathogenic in isotypes linked with neurodevelopmental phenotypes, and lower than -4.83 for isotypes linked with other phenotypes.

## The most poorly predicted tubulin pathogenic mutations likely act *via* non-loss-of-function mechanisms

To further examine the contributing factors behind the limited performance of VEPs in detecting pathogenic tubulin mutations, we sought to identify the pathogenic mutations predicted most poorly by current approaches. To do this, we developed a ranking method to examine mutations across all the VEPs used in this study. First, all predictions of gnomAD and pathogenic tubulin mutations were amalgamated. We then transformed all scores from individual VEPs to be on the same scale for comparison. Next, for each VEP, cumulative distribution ranks were computed for normalisation within a window partition across the combined dataset. Finally, for each mutation, these normalised ranks were averaged across all VEPs to generate a new metric we termed mean cumulative distribution (MCD), provided in S5 Table. Essentially the MCD provides a single value for each mutation representing how damaging it is predicted to be across all of the VEPs we used in our analysis, ranging from 0 for mildest to

1 for most disruptive. MCD replicated the results we observed for the VEP analysis, with the most significant differences between pathogenic and gnomAD values being observed in the same isotypes (TUBβ2B, TUBβ3, TUBβ4A) that showed higher ROC values (Fig F in S1 Text).

Next, we used the MCD scores to identify pathogenic tubulin mutations that were most poorly predicted by the VEPs (Table 2). These mutations generally involve substitutions between chemically similar amino acids. In particular, mutations from valine to isoleucine at position 353 in three different isotypes–TUBα1A, TUBβ5 and TUBβ8 –are among the worst predicted, with the variant in TUBβ8 having the lowest MCD score of any pathogenic tubulin mutation. Phenotypically, in TUBα1A and TUBβ5, V353I is related to the malformations of cortical development [15], while the mutation is associated with female infertility in TUBβ8, although whether the mutation is causing the issue is not shown [50].

Consequently, we experimentally tested whether mutations in the 353 position for TUBα1 and TUBβ8 enabled tubulin incorporation into microtubules. As V353I introduces a mild physicochemical change, as with most of the poorly predicted pathogenic tubulin mutations, we also tested a more disruptive mutation in both isotypes where the hydrophobic valine is changed to a positively charged arginine. V353R has to our knowledge not been observed in humans (nor is it possible with only a single nucleotide substitution), but it allows us to compare an apparently mild substitution to a more disruptive substitution at the same position. We transiently transfected the fluorescently tagged tubulin mutants into HeLa cells and analysed whether they were incorporated into the mitotic spindle, where microtubules can be clearly observed and where spindle architecture relies on MAPs. N-terminally tagged mCherry-TUBα1 incorporated into microtubules with the aid of a linker; however mCherry-TUBβ8 did not. We therefore tested a C-terminally tagged TUBβ8-GFP, which did successfully incorporate.

**Table 2. The most poorly predicted pathogenic tubulin mutations.** For each of the VEPs used in this study, cumulative distribution ranks were generated in a combined dataset of putatively benign and pathogenic tubulin variants. MCD scores for each variant were calculated by averaging the ranks from each predictor. Pathogenic variants with the lowest MCD scores are shown here, representing those most poorly predicted by VEPs.

| Isotype | Mutation | MCD Score |
|---------|----------|-----------|
| TUBß8 | V353I | 0.188 |
| TUBa1A | I238V | 0.215 |
| TUBa1A | I219V | 0.215 |
| TUBß8 | I4L | 0.221 |
| TUBa1A | D127E | 0.245 |
| TUBß8 | A352S | 0.247 |
| TUBß1 | R307H | 0.253 |
| TUBß8 | M330I | 0.280 |
| TUBß5 | V353I | 0.280 |
| TUBa1A | S54N | 0.300 |
| TUBß8 | R2K | 0.311 |
| TUBa1A | A270S | 0.326 |
| TUBa1A | V353I | 0.335 |
| TUBa1A | I188L | 0.341 |
| TUBß3 | M323V | 0.342 |
| TUBß8 | F200L | 0.352 |
| TUBß8 | V175M | 0.354 |
| TUBa1A | I5L | 0.357 |
| TUBa1A | V409I | 0.371 |
| TUBa1A | R214H | 0.372 |

Both wild-type TUBα1 and TUBβ8 isotypes strongly incorporated into the mitotic spindle in HeLa cells, co-localising with tubulin (Fig 6A and 6B and 6G in S1 Text). The V353I mutants also showed incorporation that was not significantly different from wild type for both isotypes. This is also consistent with previous work showing that V353I in TUBβ5 can still form a heterodimer and be incorporated into microtubules [28]. Interestingly, however, the V353R mutant showed severe reduction in incorporation for TUBα1, but no significant effect for TUBβ8. We also tested whether the tubulin mutations affected mitotic spindle morphology and measured the mitotic spindle length (Fig 6C). Notably, while TUBα1 V353I did not show reduced incorporation into microtubules, it did lead to significantly shorter spindles, suggesting the mutation may ultimately interfere with correct microtubule dynamics or interactions with microtubule-associated proteins. Other mutants showed no significant difference from wild type. We also confirmed that spindle aspect ratios were similar between wild-type and mutant tubulin constructs, indicating spindle scaling is conserved (Fig H in S1 Text).

Our results clearly suggest a possible molecular mechanism underlying genetic disease in the TUBα1A V353I mutation: the mutant can incorporate into microtubules along with wild-type protein, but appears to have a dominant-negative effect on spindle organisation of microtubules. In contrast, the V353R mutation, which we predict to be more severe at a protein structural level, shows reduced incorporation and thus has no effect on spindle morphology. This mirrors the recent observation that dominant-negative and gain-of-function mutations tend to be more structurally mild than loss-of-function mutations [37].

None of the tested TUBβ8 mutations show any phenotypic effect. It is interesting to note that V353 residues in tubulin-α and -β are not at precisely equivalent positions in a sequence alignment or in the structures of tubulin-α and -β (Fig I in S1 Text). V353 is a surface residue in tubulin-α, close to the longitudinal interdimeric interface, whereas in tubulin-β, it is more buried in the protein interior, but close to the intradimer interface, and thus it is unsurprising that mutations at V353 could have different molecular effects in each isotype. It is possible that the molecular effects of V353 mutations in TUBβ8 are too subtle to detect in our experimental system. Alternatively, the evidence that V353I is a true pathogenic mutation in TUBβ8 is quite limited. It was observed in a heterozygous state in a single patient with infertility, but it is unknown if either of her parents also had the mutation [50]. Thus, another possible explanation for the poor computational prediction of TUBβ8 V353I is that it is not really pathogenic.

## Discussion

Tubulinopathies have previously been associated with defects in cortical brain development [15,57,63–66]. However, several pathogenic tubulin mutations reported in recent years are not linked to neurodevelopmental diseases, and thus the phenotypical profile of 'tubulinopathies' needs redefining. As the mechanisms by which tubulin mutations give rise to these disease phenotypes are generally unknown, the extensive repertoire of newly identified mutations now at our disposal provides an opportunity to study pathogenicity globally, across the tubulin family.

The strong conservation in tubulin sequence and structure across isotypes [12], and the lethality of tubulin mutations in yeast [67] might lead to the assumption that many pathogenic mutations would cause protein instability and consequently a loss of function. However, this appears not to be the case, with predictions of changes in protein stability showing very little ability to discriminate between pathogenic and putatively benign mutations, with the exception of a slight trend observed for TUBβ2B. There is some previous evidence for a loss-of-function mechanism underlying some pathogenic tubulin mutations, e.g. the R264C mutation in TUBα1A perturbs chaperone binding and reduces the extent of heterodimer assembly, which

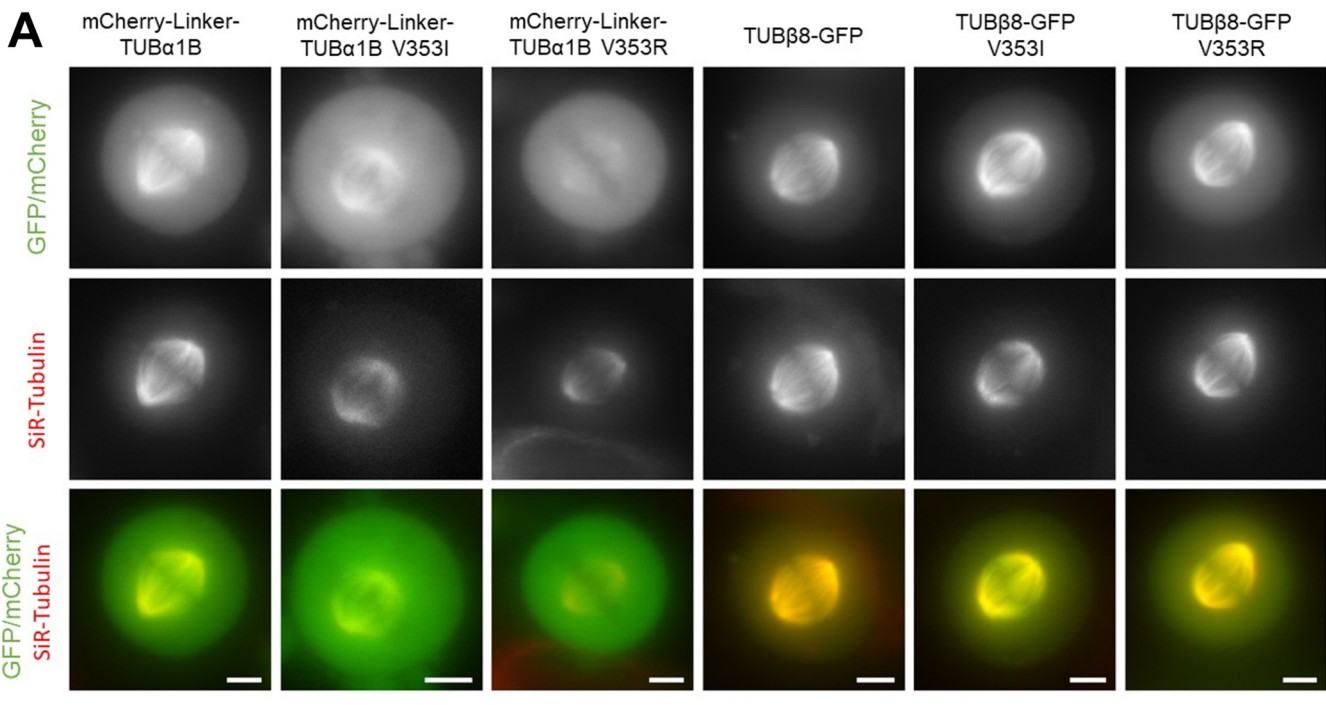

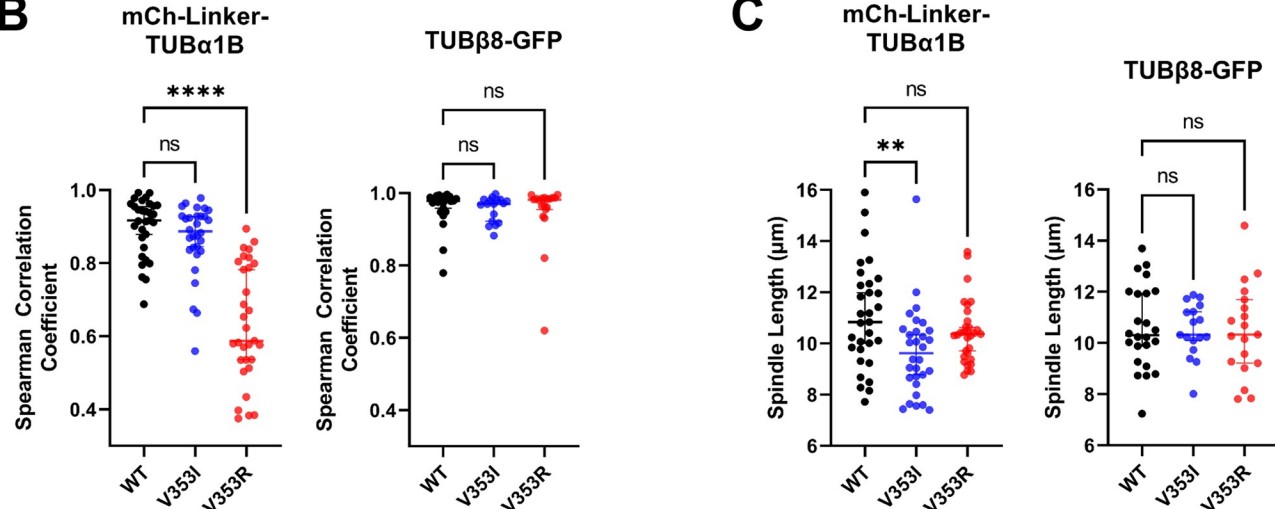

**Fig 6. Effects of V353 mutants on microtubules and mitotic spindles in cells. (A)** Representative live-cell images of mitotic HeLa cells after being transfected with fluorescently tagged wild type and mutant tubulin-α1B and tubulin-β8 constructs (green) and incubated with SiR-tubulin (red) **(B)** Wide linescans across the whole mitotic spindle were taken to measure fluorescence intensity of tubulin across the mitotic spindle compared to total tubulin. For each cell, two-sided Spearman correlations were calculated between both signal intensities. **(C)** Spindle length measurements for the cells transfected with fluorescently tagged wild-type and mutant tubulin-α1B and tubulin-β8 and measured in (B). For each construct, median values are displayed with 95% confidence intervals. Asterisks indicate Kruskal-Wallis test (with post-hoc Dunn) significance values, ***, $P < 0.001$; ****, $P < 0.0001$. Bars, 5μm.

has been suggested to account for its pathogenic effects [25]. However, it appears that the overall role of destabilisation in driving tubulin pathogenicity must be very limited.

Tubulin mutations can act via dominant-negative or gain-of-function mechanisms, where the mutant protein incorporates into the microtubule lattice and instead disrupts its function.

These effects have been reported in tubulins before and include mutations in TUBα1A [27] and others across different tubulin-β isotypes [28–32]. Once they are incorporated into microtubules, these mutations may alter biophysical microtubule properties or disrupt interactions with MAPs. The latter model includes interactions with non-motor MAPs, like NuMA and PRC1 [31,32], and the impairment of motor protein binding and, subsequently, cellular transport [33]. The disruption of tubulin-MAP interactions may be an especially pertinent mechanism given that mutations on protein-protein interfaces are more likely to be pathogenic [68]. Here, multiple copies of mutant tubulin incorporated into microtubule can compound a mutation's effect. Also, some tissues express multiple tubulin-β isotypes, especially in the developing brain [69]. Here, redundancy may mitigate any loss-of-function in a specific isotype, but cells struggle to cope with dominant-negative or gain-of-function effects.

The classic definition of the dominant-negative effect involves the mutant protein disrupting the activity of the wild-type protein, either directly or indirectly, whereas the phenotypic effects of gain-of-function mutations are due to the mutant protein doing something different than the wild-type protein [26]. For most proteins that form relatively small complexes, classification of a mutation as being associated with to either mechanism should usually be straightforward. However, given the large size of microtubule assemblies and their diverse functional roles, such classification may be ambiguous for tubulins. It depends on whether disease is due to reduced microtubule function (dominant negative), or whether altered microtubule properties are driving disease (gain of function). While here we have referred to TUBα1 V353I, which incorporates into microtubules and reduces spindle length, as dominant negative, it is possible that its pathogenic effects might be better described as gain of function. These assays give limited data because of their lack of sensitivity. Further experimental work and more quantitative assays are required to understand the effect of these types of tubulin mutations in the cellular and molecular context.

The pathogenic mutations predicted most poorly tend to involve substitutions between similar amino acids, with hydrophobicity and charges largely maintained between wild-type and mutant residues. This observation likely reflects the fact that most predictors incorporate some manner of amino acid substation matrix, thus predicting more minor effects for variants that do not change a residue's charge or hydrophobic state. In contrast to more drastic mutations that tend to be better predicted, these mutations may be less detrimental to folding and microtubule assembly. Indeed, this difference is likely to explain why we observed a weaker co-localisation to the mitotic spindle in the TUBα1 V353R mutant compared to V353I.

In this study, the performance of supervised predictors, and especially metapredictors like REVEL and M-CAP, highlights the issue of circularity present in these types of VEPs, which are typically trained using published pathogenic and benign variants [70]. There is a strong likelihood that some of these mutations are in training sets for these predictors. These biases would lead metapredictors and supervised methods to predict mutations in isotypes linked with neurodevelopmental diseases as pathogenic correctly, while mutations in other isotypes would be predicted poorly. This may explain why metapredictors performed so much worse on such mutations, especially when compared to unsupervised methods like DeepSequence and PROVEAN.

We judge DeepSequence to be the best performing VEP, considering consistency across isotypes and phenotypes. Unlike supervised methods, it does not use a labelled training set. DeepSequence can also make predictions based on the evolutionary conservation of entire sequences at once as opposed to no more than a few sites of interest, setting it apart from many other VEPs [65]. Benchmarking several VEPs against deep mutational scanning data also highlighted DeepSequence as the best pathogenicity predictor [62]. We, therefore, recommend using DeepSequence over metapredictors like REVEL and M-CAP to achieve more accurate predictions of tubulin variant pathogenicity.

This study presents the tubulin family as a blind spot for current phenotype predictors, especially in identifying mutations that result in non-neurodevelopmental pathogenic phenotypes. Predictor biases due to their training sets introduce circularity and could limit predictions. The diversity of pathogenic phenotypes necessitates further study to understand the molecular mechanisms underlying their pathogenicity. Experimental approaches to dissect the effect of these tubulin pathogenic mutations on microtubule dynamics, microtubule assembly and microtubule interactions with associated proteins would contribute to a better molecular understanding of tubulinopathies [71–73]. These additional data could refine current phenotype predictors and increase their performance.

For non-neurodevelopmental phenotypes, pathogenic effects initially appeared to be isotype-specific, but this could be linked to where these isotypes are expressed and to what level, or other unknown factors. For example, pathogenic mutations in TUBβ8 are associated to female infertility, but males carrying these mutations would presumably present a healthy phenotype [22]. Importantly, many tubulin mutations are currently reported as linked to a disease or phenotype, but the evidence supporting causality is limited. It will be important to demonstrate whether the mutation is causative of pathogenesis or a consequence of other changes to cell homeostasis or genome stability. A pragmatic experimental approach is therefore essential to define the mechanism of pathogenesis associated with tubulin mutations. It will then be possible to develop strategies to treat tubulin-related disorders using a personalised medicine approach.

## Methods

### Mutation datasets

A thorough literature search was first performed to identify as many pathogenic or likely pathogenic missense mutations across the tubulin family as possible. While the recently released tubulin mutation database [39]–as well as others such as OMIM [74] and ClinVar [38]–were excellent starting points, mutations from other publications (especially ones reported recently) were included (all detailed in Table 1). The literature search also uncovered a few other mutations that were incompletely dominant, homozygous, or of an unknown heritability, but these were omitted from the study. Any mutations described as *de novo* were assumed to be heterozygous and hence also dominant. As the underlying mechanisms with pathogenic mutations leading to cancer are likely to be more complex (and distinct), somatic mutations linked with such phenotypes were excluded. A set of putatively benign missense mutations for each isotype was retrieved from gnomAD v2.1 [40]. We excluded mutations present in both pathogenic and gnomAD datasets and considered only isotypes with more than one pathogenic mutation.

### Protein structural analyses

Three-dimensional models of tubulin structures were taken from the protein data bank (PDB) on 2020.05.27, using the first biological assembly from each entry to represent its quaternary structure. We searched for polypeptide chains with at least 70% sequence identity to human tubulin isotype over a stretch of at least 50 amino acid residues. This allowed structural analysis using related structures for some isotypes for which structures are not available. Importantly, we only considered those residues where the amino acid in the structure, as well as both adjacent residues, are the same as in the sequence of the isotype of interest. Unless otherwise stated, all structures that satisfy this threshold were considered. A hierarchy was implemented to determine where a residue occurred on the tubulin structure concerning its function (Fig B in S1 Text). First, we looked through all structures to highlight residues interacting with GTP (or its analogues). Then, model structures were used (S2 Table) to determine whether the

remaining residues occur on interfaces between the corresponding subunit in the tubulin dimer or with MAPs and other tubulin dimers in the microtubule in that specified order. Residues not present on any interface were mapped to the highest resolution structure within the dataset. These mappings were used to classify residues according to their location, separating residues on the surface (relative solvent accessible surface area >0.25) from ones at the protein interior. Once every residue was labelled, their distribution was compared between mutations in our pathogenic and gnomAD datasets. Finally, all mutations in tubulin-α or -β were considered before moving on to specific isotypes with at least 5 identified pathogenic missense mutations, also using PyMOL [75] to confirm that these patterns could also be observed qualitatively. To facilitate further analysis of tubulin mutations, we have provided a list of all amino acids in tubulin-α and- β that are involved in protein-protein or protein-ligand interactions (S6 Table).

Structural predictions were then obtained from FoldX [60] for pathogenic and gnomAD variants. Each mutation was mapped onto the highest resolution structure from the dataset above. Here, the ΔΔG values were calculated considering the monomer only, as well as the whole biological assembly, in the same manner as previously described [37]. All FoldX predicted ΔΔG values are provided in S7 Table.

## Computational variant effect predictions

Scores from VEPs were generated using the same pipeline as described previously [62]. Though most of the methods used failed to produce scores for the entire combined dataset, the percentage of missing scores from each predictor did not exceed 7.8%, except for S3D-PROF and SNPs&GO-3D, which required PDB structures as inputs. For S3D-PROF and SNPs&GO-3D, structures with the closest sequence identity were used, which have the following PDB IDs: 5iyz (TUBα1A), 5lxt (TUBα4A), 4tv9 (TUBβ1), 5nm5 (TUBβ2A and TUBβ8), 4i4t (TUBβ2B), 4lnu (TUBβ3), 5jqg (TUBβ4A and TUBβ4B) and 4zol (TUBβ5).

For plotting ROC curves, mutations from the pathogenic dataset were labelled as true positives, with gnomAD variants marked as true negatives. The plots themselves, as well as the AUC, were calculated using the PRROC package in R. Due to predictors PROVEAN, DeepSequence, SIFT4G and BLOSUM operating via inverse metrics (i.e. variants predicted to be more damaging have lower scores), all scores were multiplied by -1 before plotting curves and calculating AUC values. Finally, ROC AUC values were calculated globally across the whole dataset for each VEP first before generating specific scores of each VEP for individual isotypes.

To calculate the MCD rankings, the datasets containing prediction scores for gnomAD and pathogenic mutations were combined. Then, the cumulative distribution function was applied to the collective scores obtained from each predictor using the cume_dist function from the DPLYR package on a per-VEP basis. Each value represents the proportion of scores that are less than or equal to the one obtained for that particular mutation from a specific VEP. Then, the cumulative distribution scores from all VEPs were averaged for each mutation to generate the MCD score.

## Molecular cloning

The tubulin genes human TUBα1B (NM_006082.3) or TUBβ8 Class VIII (NM_177987.2) were synthesised by Genewiz. The TUBα1B gene were cloned into pBABE- blasticidin containing an N-terminal mCherry tag [76]. An additional linker region (with sequence 5'-ATGAGGAGGGGCGCTGCCGATAGGGAAACTGAGAGGCTCCCCGGCGCA-CAAGGTCCGTGCAGTGCGGTCAGTGCGGCCAGCTCCACATTGGCCGCAG-

CAGCGGCCCCTCGTGCTCGGGCGACCGCTGCCGCGTCCACCCTCAGCGC-CACCGCCCTCGAG-3') was also inserted between the N-terminal mCherry tag and the start of the tubulin sequence. The TUBβ8 gene was cloned into a pBabe-blasticidin containing a C-terminal GFP. Mutagenesis was performed using the Agilent Quickchange II mutagenesis kit, as per the manufacturer's instructions. While the TUBα1B sequence was used rather than TUBα1A, where the V353I disease mutations has been identified, these are nearly identical, with only two minor amino acid sequence differences between them (G versus S at residue 232 and T versus S at residue 340), neither being close to V353 in the three-dimensional structure.

## Cell biology and fluorescence imaging

HeLa cells were cultured and maintained in a humidified 5% $CO_2$ atmosphere at 37°C in DMEM supplemented with 100 U/ml penicillin and 100 μg/ml streptomycin (Gibco) and 10% Fetal Bovine Serum (Gibco). Cells were plated on 35 mm glass-bottom microwell dishes (ibidi) for live-cell imaging. Lipofectamine 3000 reagent (Invitrogen) was used for transient transfections according to manufacturer guidelines. 24–48 hours after transfection, cells were incubated with 20–50 nM SiR-tubulin dye (SpiroChrome) for 1–2 hours and transferred to L-15 medium (ThermoFisher Scientific) before imaging. Images were taken using a Nikon Ti2 live imaging microscope were stored on OMERO [77], which was used to manually measure spindle length and width. ImageJ was then used to process the acquired images [78]. A region of interest was drawn over the spindle length observed through the Cy5 channel (visualising SiR-tubulin), and intensity profiles were generated for this channel and the GFP or mCherry channel (depending on the construct). These profiles were then background subtracted using the average of the intensity profiles of two thinner lines on either side of the spindle. For each cell, a two-sided Spearman's correlation coefficient was calculated between the fluorescent and SiR-tubulin intensity profile. OMERO was also used to calculate spindle length and width, where the length was divided by width to obtain calculate the spindle aspect ratio. Plots were visualised using GraphPad Prism 9, which was also used to carry out Kruskal-Wallis tests (with post-hoc Dunn).

## Supporting information

**S1 Text. Supplemental figures.** File containing additional figures complementary to the analysis shown in the main text of this manuscript.
(DOCX)

**S1 Table. List of tubulin mutations.** Full description of all pathogenic mutations compiled in this study (including phenotypes and references) as well as a list of all gnomAD mutations used.
(XLSX)

**S2 Table. Structures used for location assignments.** All model structures used for the hierarchy implemented for assigning structural locations for tubulin residues.
(CSV)

**S3 Table. Variant effect predictions.** Complete set of raw predictions obtained from all the variant effect predictors used in this study for every tubulin mutation.
(CSV)

**S4 Table. DeepSequence outputs.** DeepSequence predictions for every possible missense mutation for all tubulin isotypes considered in this study.
(CSV)

**S5 Table. Mean cumulative distribution scores.** Ranking of mean cumulative distribution (MCD) scores for all pathogenic tubulin mutations.
(CSV)

**S6 Table. Tubulin residue interactions.** All tubulin residues found to be interacting with GTP (or its analogues), MAPs, or with other tubulin subunits (interdimer or intradimer).
(CSV)

**S7 Table. Predicted effects on protein stability for tubulin mutations.** All ΔΔG values predicted using FoldX using both the structure of the monomeric subunit (ddG_fold) and the full biological assembly (ddG_full).
(CSV)

**S1 Data. Multiple sequence alignment.** Multiple sequence alignment of all tubulin-α and β isotypes considered in this study.
(PDF)

## Acknowledgments

We thank Benjamin Livesey for help with generating the VEP outputs.

## Author Contributions

**Conceptualization:** Julie P. I. Welburn, Joseph A. Marsh.

**Data curation:** Joseph A. Marsh.

**Formal analysis:** Thomas J. Attard.

**Investigation:** Thomas J. Attard.

**Supervision:** Julie P. I. Welburn, Joseph A. Marsh.

**Writing – original draft:** Thomas J. Attard.

**Writing – review & editing:** Julie P. I. Welburn, Joseph A. Marsh.

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
