## [Decision Letter · Decision Letter 0]

13 Sep 2022

Dear Marsh,

Thank you very much for submitting your manuscript "Understanding molecular mechanisms and predicting phenotypic effects of pathogenic tubulin mutations" for consideration at PLOS Computational Biology. As with all papers reviewed by the journal, your manuscript was reviewed by members of the editorial board and by several independent reviewers. The reviewers appreciated the attention to an important topic. Based on the reviews, we are likely to accept this manuscript for publication, providing that you modify the manuscript according to the review recommendations.

Sincerely,

Bert L. de Groot

Academic Editor

PLOS Computational Biology

Arne Elofsson

Section Editor

PLOS Computational Biology

[LINK]

Reviewer's Responses to Questions

**Comments to the Authors:**

Reviewer #1: Alpha Tubulin and beta tubulin, which make up microtubules, have been linked to a variety of disorders that are often dominant, sporadic, and congenital. Although the first identified tubulin mutations are involved with neurodevelopment, mutations are also linked to other conditions, such as bleeding disorders and infertility. The authors observe that the co-localization of few tubulin isotype mutations with the mitotic spindle in HeLa cells suggests that they may exhibit dominant-negative effects through affecting microtubule characteristics. The authors suggest that they constitute a blind area for existing computational methods, since they are less accurately predicted than mutations in the majority of human diseases. Their computational supporting data supports their conclusions. There have been very few computational studies conducted on tubulin isotypes and their function. This work provide some novel insights on tubulin mutations and their role in pathogenicity which could be further tested in future through experiments.

Reviewer #2: See the attached file.

**Have the authors made all data and (if applicable) computational code underlying the findings in their manuscript fully available?**

Reviewer #1: Yes

Reviewer #2: None

PLOS authors have the option to publish the peer review history of their article (what does this mean?). If published, this will include your full peer review and any attached files.

Reviewer #1: No

Reviewer #2: No

Figure Files:

Data Requirements:

Reproducibility:

References:

---

## [Editor Report · Decision Letter 1]

28 Sep 2022

Dear Marsh,

We are pleased to inform you that your manuscript 'Understanding molecular mechanisms and predicting phenotypic effects of pathogenic tubulin mutations' has been provisionally accepted for publication in PLOS Computational Biology.

Best regards,

Bert L. de Groot

Academic Editor

PLOS Computational Biology

Arne Elofsson

Section Editor

PLOS Computational Biology

---

## [Editor Report · Acceptance letter]

3 Oct 2022

PCOMPBIOL-D-22-00932R1 

Understanding molecular mechanisms and predicting phenotypic effects of pathogenic tubulin mutations

Dear Dr Marsh,

I am pleased to inform you that your manuscript has been formally accepted for publication in PLOS Computational Biology. Your manuscript is now with our production department and you will be notified of the publication date in due course.

With kind regards,

Zsofia Freund
